

# voomDDA: discovery of diagnostic biomarkers and classification of RNA-seq data

Gokmen Zararsiz[1,2], Dincer Goksuluk[2,3], Bernd Klaus[4], Selcuk Korkmaz[2,5], Vahap Eldem[6], Erdem Karabulut[3] and Ahmet Ozturk[1,2]

[1] Department of Biostatistics, Erciyes University, Kayseri, Turkey
[2] Turcosa Analytics Solutions Ltd Co, Erciyes Teknopark 5, Kayseri, Turkey
[3] Department of Biostatistics, Hacettepe University, Ankara, Turkey
[4] European Molecular Biology Laboratory, Heidelberg, Germany
[5] Department of Biostatistics, Trakya University, Edirne, Turkey
[6] Department of Biology, Istanbul University, Istanbul, Turkey

Corresponding author
Gokmen Zararsiz,
gokmenzararsiz@hotmail.com,
gokmenzararsiz@erciyes.edu.tr

## ABSTRACT

RNA-Seq is a recent and efficient technique that uses the capabilities of next-generation sequencing technology for characterizing and quantifying transcriptomes. One important task using gene-expression data is to identify a small subset of genes that can be used to build diagnostic classifiers particularly for cancer diseases. Microarray based classifiers are not directly applicable to RNA-Seq data due to its discrete nature. Overdispersion is another problem that requires careful modeling of mean and variance relationship of the RNA-Seq data. In this study, we present voomDDA classifiers: variance modeling at the observational level (voom) extensions of the nearest shrunken centroids (NSC) and the diagonal discriminant classifiers. VoomNSC is one of these classifiers and brings voom and NSC approaches together for the purpose of gene-expression based classification. For this purpose, we propose weighted statistics and put these weighted statistics into the NSC algorithm. The VoomNSC is a sparse classifier that models the mean-variance relationship using the voom method and incorporates voom's precision weights into the NSC classifier via weighted statistics. A comprehensive simulation study was designed and four real datasets are used for performance assessment. The overall results indicate that voomNSC performs as the sparsest classifier. It also provides the most accurate results together with power-transformed Poisson linear discriminant analysis, rlog transformed support vector machines and random forests algorithms. In addition to prediction purposes, the voomNSC classifier can be used to identify the potential diagnostic biomarkers for a condition of interest. Through this work, statistical learning methods proposed for microarrays can be reused for RNA-Seq data. An interactive web application is freely available at http://www.biosoft.hacettepe.edu.tr/voomDDA/.

## INTRODUCTION

In molecular biological studies, gene-expression profiling is among the most widely applied genomics techniques to understand the role and the molecular mechanism of particular genes in conditions of interest (*Law et al., 2014*). Recent high-throughput technologies allow researchers to quantify the expression levels of thousands of genes simultaneously. During the last two decades, microarray technology was very popular in gene expression profiling. Due to several major advantages, RNA-Seq technology replaced microarrays as the technology of choice and become the de facto standard in gene-expression studies (*Ritchie et al., 2015*).

Identifying the relevant genes across the conditions (e.g., tumor and non-tumor tissue samples) is a common research interest in gene-expression studies. One major task is to detect the minimal set of genes which gives the maximum predictive performance for the diagnostic purpose of samples in medicine. A particular interest is the cancer classification based on the simultaneous monitoring of thousands of genes (*Díaz-Uriarte & De Andrés, 2006*).

For microarray studies, a great deal of machine learning algorithms have been proposed and applied for gene-expression based classification. However, these algorithms cannot be directly applied to RNA-Seq data, since the type of the data is entirely different. In contrast to the continuous data format of microarrays, RNA-Seq data are summarized with nonnegative and integer-valued counts, which are obtained from the number of mapped sequencing reads to genomic regions of the species of interest.

For the classification purpose, there is still less advancements for RNA-Seq data until recently. *Witten (2011)* proposed a Poisson linear discriminant analysis (PLDA) classifier, which is an extension of Fisher's linear discriminant analysis to high-dimensional count data. PLDA shrinks the class differences to identify a subset of genes and applies a Poisson log linear model for classification (*Witten, 2011*). *Dong et al. (2015)* extended this algorithm to build a new classification method based on the negative binomial (NB) distribution. The authors used a shrinkage method to predict the additional overdispersion parameter. Another solution may be to transform the count data into the continuous format to bring RNA-Seq data hierarchically closer to the microarray data and make use of the flexibility of normal distribution.

Recently, variance modeling at the observational level (voom) method has been proposed to open access microarray based methods for RNA-Seq analysis (*Law et al., 2014*). The voom method estimates the mean and the variance relationship from the log counts and provides precision weights for downstream analysis. This method is integrated with the limma (linear models for microarray and RNA-Seq data) method (*Ritchie et al., 2015*) and showed the best performance as compared to count based methods in controlling the type-I error rate, having the best power and lowest false discovery rate. The clear advantages of voom over other methods and its good integration with limma for differential expression analysis may point to high predictive performance in classification and clustering tasks. Despite these advantages, the voom method has only been used for differential-expression studies. There are no studies in the literature that use the voom method for classification purposes.

In this paper, we introduce voomNSC sparse classifier, which brings two powerful methods together for the classification of RNA-Seq data: voom and the nearest shrunken centroids (NSC) algorithm (*Tibshirani et al., 2002*). For this purpose, we propose weighted statistics and adapt the NSC algorithm with these weighted statistics. Basically, voomNSC accepts either a normalized or non-normalized count data as input, applies voom method to data, provides precision weights for each observation and ultimately, fits an adapted NSC classifier by taking these weights into account. Thus, the main objective of proposing this method is twofold:

1. to extend voom method for RNA-Seq classification studies;
2. to make NSC algorithm available for RNA-Seq technology.

We also made available the diagonal discriminant classifiers (*Dudoit, Fridlyand & Speed, 2002*) to be able to work with RNA-Seq data. Two diagonal RNA-Seq discriminant classifiers, voomDLDA and voomDQDA, will also be presented within the scope of this study. All three classifiers will be referred as voomDDA classifiers throughout this paper.

We organized the rest of this study as follows. In the 'Materials & Methods' section, we present the underlying theory of voomDDA classifiers and detail the experiments. In the 'Results' section, we give the results of simulation and real dataset experiments. We discuss and conclude our study in the 'Discussion' and 'Conclusion' sections. Extended information about the methods background, experiments, results and software source codes are available in Files S1–S4.

## MATERIALS & METHODS

### VoomDDA classifiers

In this section, we detail the methodology of voomDDA classifiers. We assume that the input data is a $p \times n$ dimensional count data matrix, where $p$ refers to the number of features and $n$ refers to the number of samples. Input data may consist of either $x_{gi}$ raw or $x'_{gi}$ normalized count values. Moreover, genes with zero or very low counts should be filtered before starting the analysis. For simplicity, we will assume throughout this section that the input data, $X$, is a $p \times n$ dimensional, filtered and non-normalized count data matrix.

### Calculation of log-cpm values and estimation of precision weights

Firstly, we get the voom estimates, i.e., log-counts and associated precision weights, as described in *Law et al. (2014)*. Let $X_{.i}$ be the library size for sample $i$. We start by calculating the log-counts-per million (log-cpm) values using the Eq. (1):

$$z_{gi} = \log_2 \left( \frac{x_{gi} + 0.5}{X_{.i} + 1} \times 10^6 \right). \tag{1}$$

Small constant values 0.5 and 1 in the formula are used to avoid taking the logarithm of zero and guaranteeing that $0 < (x_{gi} + 0.5)/(X_{.i} + 1) < 1$. To estimate the precision weights $w_{gi}$, we take advantage of the delta rule, linear models and lowess smoothing curves. We assume a linear model between the expected size of the log-cpm values and the class conditions as follows:

$$E(z_{gi}) = \mu_{z_{gi}} = y_i^T \beta_g. \tag{2}$$

In the formula, $\beta_g$ corresponds to a vector of regression coefficients to be estimated. These coefficients are the log-fold-changes between class conditions (*Law et al., 2014*). Matrix notation of this equation is as follows:

$$E(z_g) = D\beta_g \tag{3}$$

where **D** represents the design matrix with the rows $y_i$ and $z_g$ is a vector containing the log-cpm values for $g$th gene. For each gene, we fit these models using ordinary least squares method and obtain the fitted coefficient $\hat{\beta}_g$, the fitted log-cpm values, $\hat{\mu}_{z_{gi}} = y_i^T \hat{\beta}_g$, and the standard deviations of residuals $s_g$.

Let $\bar{z}_g = \sum_{i=1}^n \hat{\mu}_{z_{gi}}/n$ be the mean log-cpm value for $g$th gene, and $\tilde{X}_{.n} = \left(\prod_{i=1}^n (X_{.i}+1)\right)^{1/n}$ be the geometric mean of the library sizes plus one. Using delta rule, we obtain the mean log-counts $\tilde{x}_g$ as follows:

$$\tilde{x}_g \approx \bar{z}_g + \log_2(\tilde{X}_{.n}) - 6\log_2(10). \tag{4}$$

Log counts are calculated from the fitted log-cpm values accordingly:

$$\hat{\mu}_{gi} \approx \hat{\mu}_{z_{gi}} + \log_2(X_{.i}+1) - 6\log_2(10). \tag{5}$$

Now, we estimate the mean–variance relationship for each gene, using the mean log counts $\tilde{x}_g$ and the square root of residual standard deviations $s_g^{1/2}$. A lowess curve (*Cleveland, 1979*) is fitted using the smoothing function $g(.)$ as follows:

$$s_g^{1/2} = g(\tilde{x}_g). \tag{6}$$

A piecewise linear function $lo(.)$ is obtained from the fitted lowess curve by interpolating the curve for the $\tilde{x}_g$ values in order. Finally, we obtain the $w_{gi}$ precision weights (i.e., inverse variances of log-cpm values) as follows:

$$w_{gi} = lo(\hat{\mu}_{gi})^{-4}. \tag{7}$$

The log-cpm values, $z_{gi}$, and the associated precision weights, $w_{gi}$, will be used in the model building process of voomDDA classifiers.

## Classification models based on diagonal weighted sample covariance matrices

First of all, we assume that genes are independent of each other in building classification rules. Let $i_k, \ldots, i_{k+1} - 1$ belong to class $k$, $k \in 1, \ldots, K$, $n_k$ is the number of samples in class $k$ and we set $i_{K+1} = n + 1$. Let $\bar{z}_{w_{gk}} = \left(\sum_{i=i_k}^{i_{k+1}-1} w_{gi}z_{ik}\right)/\sum_{i=i_k}^{i_{k+1}-1} w_{gi}$ be the class-specific weighted mean for $k$th class, $\bar{z}_{w_g} = \left(\sum_{k=1}^K n_k \bar{z}_{w_{gk}}/n\right)$ be the overall weighted mean, $\hat{\Sigma}_{wC=k} = diag\left(s_{w_{1k}}^2, \ldots, s_{w_{pk}}^2\right)$ be the diagonal weighted sample covariance matrices for $k$th class and $\hat{\Sigma}_w = diag\left(s_{w_1}^2, \ldots, s_{w_p}^2\right)$ be the weighted pooled covariance matrix. The diagonal elements of these matrices are obtained from the class specific and the pooled weighted variances respectively. The off-diagonal elements of these matrices are all set to zero. The weighted pooled variance of $g$th gene can be calculated as follows:

$$s_{w_g}^2 = \sum_{i=1}^K (n_k - 1)s_{w_{gk}}^2/(n - K). \tag{8}$$

The weighted variance of $g$th gene in class $k$ can be calculated as follows:

$$s_{w_{gk}}^2 = \frac{\sum_{i=i_k}^{i_{k+1}-1} w_{gi}}{\left(\sum_{i=i_k}^{i_{k+1}-1} w_{gi}\right)^2 - \sum_{i=i_k}^{i_{k+1}-1} w_{gi}^2} \sum_{i=i_k}^{i_{k+1}-1} w_{gi}\left(z_{gi} - \bar{z}_{w_{gk}}\right)^2. \tag{9}$$

Using these weighted statistics, we define voomDLDA and voomDQDA classifiers, which are extensions of DLDA and DQDA classifiers for RNA-Seq data with the weighted parameter estimates. voomDLDA assumes that the gene specific weighted variances are equal across groups and uses the weighted pooled covariance matrix in modeling class-conditional densities $f_k(x)$. On the other hand, voomDQDA uses separate covariance matrices $\hat{\Sigma}_{w_{C=k}}$, which are obtained from class-specific weighted variance statistics.

## Prediction of test observations for VoomDLDA and VoomDQDA classifiers

Discriminant rules for voomDLDA and voomDQDA classifiers are given below:

$$\delta_k^{\text{voomDLDA}}(x_*) = -\sum_{g=1}^{p} \frac{\left(z_{g*} - \bar{z}_{w_{gk}}\right)^2}{s_{w_g}^2} + 2\log(\hat{\pi}_k) \tag{10}$$

$$\delta_k^{\text{voomDQDA}}(x_*) = -\sum_{g=1}^{p} \frac{\left(z_{g*} - \bar{z}_{w_{gk}}\right)^2}{s_{w_{gk}}^2} + 2\log(\hat{\pi}_k), \tag{11}$$

where $\hat{\pi}_k$ is the prior probability (e.g., class proportions) of class $k$.

A new test observation $(x_*)$ will be assigned to one of the classes which maximizes the $\delta_k^{\text{voomDLDA}}(x_*)$ or $\delta_k^{\text{voomDQDA}}(x_*)$. An important point here is that the same parameters should be used for both training and test sets to guarantee that both sets are on the same scale and homoscedastic relative to each other. Thus, $z_{g*}$ should be obtained after normalizing and transforming $x_*$ based on the distributional parameters of the training dataset.

Suppose that the training dataset is normalized using the DESeq median ratio normalization method. Then the size factor of a test observation, $\hat{s}_*$, will be calculated as follows:

$$m_* = \text{median}_g \left\{ \frac{x_{g*}}{\left(\prod_{i=1}^n x_{gi}\right)^{1/n}} \right\} \tag{12}$$

$$\hat{s}_* = \frac{m_*}{\sqrt[n+1]{\prod_{i \in 1,\dots,n} x_{gi}}}. \tag{13}$$

If we use the trimmed mean of M values (TMM) normalization method, then a reference sample which is selected in the training set, will be used for the normalization of the test set. Let $X_{.*}$ be the library size for the test observation. Then, we calculate TMM normalization factors as follows:

$$\log_2\left(TMM_*^r\right) = \frac{\sum_{g=1}^{p'} \varpi_{g*}^r M_{g*}^r}{\sum_{g=1}^{p'} \varpi_{g*}^r} \tag{14}$$

where $M_{g*}^r = \frac{\log_2(x_{g*}/X_{.*})}{\log_2(x_{gr}/X_{.r})}$ and $\varpi_{g*}^r = \frac{X_{.*}-x_{g*}}{X_{.*}x_{g*}} + \frac{X_{.r}-x_{gr}}{X_{.r}x_{gr}}$; $x_{g*}, x_{gr} > 0$.
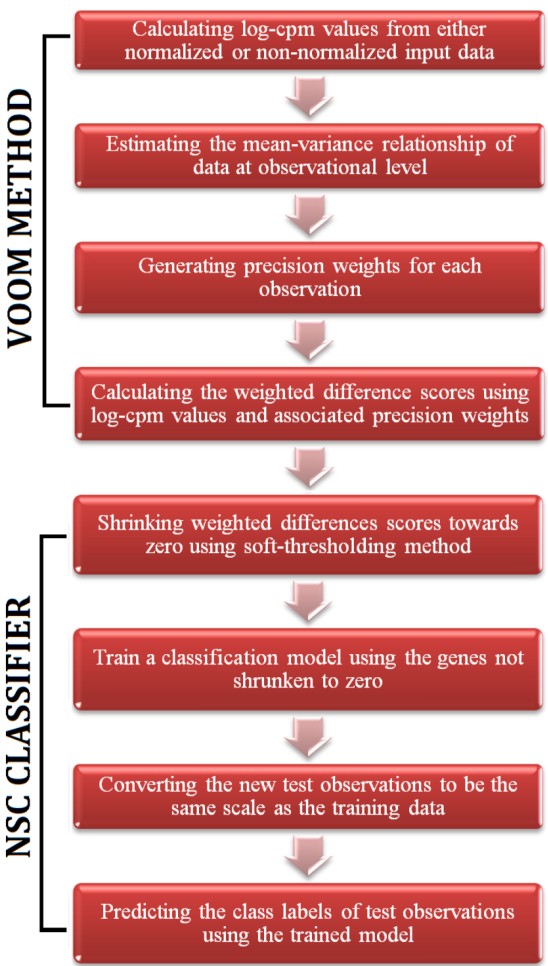

**Figure 1  A flowchart of the steps of voomNSC algorithm.**

In the voom transformation, log-cpm values for $x_*$ can be calculated as:

$$z_{g*} = \log_2\left(\frac{x_{g*} + 0.5}{X_{.*} + 1} \times 10^6\right).$$  (15)

If a normalization (e.g., DESeq median ratio, TMM, etc.) is initially applied, then the normalized values $x'_{g*} = x_{g*}/\hat{s}_*$ is used instead of $x_{g*}$ in the formula.

## Sparse VoomNSC classifier for RNA-Seq classification

The RNA-Seq data is high-dimensional just like the microarray data. Hence, one obtains very complex models from voomDLDA and voomDQDA classifiers. Here, we present the voomNSC algorithm to overcome this complexity and obtain simpler, more interpretable models with reduced variance. voomNSC, which incorporates both log-cpm values and the associated weights together into the estimation of model parameters by using the weighted statistics, is an extension of Tibshirani et al.'s (*2002, 2003*) NSC algorithm. A flowchart displaying the steps of the voomNSC algorithm is given in Fig. 1.

Like the NSC algorithm, voomNSC aims to identify the most important gene subset for class prediction. Briefly, the standardized class specific weighted gene expression means are shrunken towards the standardized overall gene expression weighted means. The genes shrunken to zero are eliminated and a voomDLDA classification model is built with the remaining genes. Mean expressions are also called centroids. Let $d_{w_{gk}}$ be the weighted difference scores, between weighted centroids of $k$th class and overall weighted centroids:

$$d_{w_{gk}} = \frac{\overline{z}_{w_{gk}} - \overline{z}_{w_g}}{m_k \left( s_{w_g} + s_{w_0} \right)} \tag{16}$$

where $m_k = \sqrt{\frac{1}{n_k} - \frac{1}{n}}$ is a standard error adjustment term and $s_{w_0}$ is a small positive constant added to the denominator of Eq. (16) to ensure that the variance of the difference scores is independent from the gene expression level. $s_{w_0}$ is calculated from the median value of $s_{w_g}$ across genes.

These weighted difference scores can be considered as the voom extension of the "relative differences" mentioned in *Tusher, Tibshirani & Chu (2000)*. One can use these scores for the purpose of differential expression analysis with the significance analysis of microarrays (SAM) method. Eq. (16) can be rewritten as in Eq. (17):

$$\overline{z}_{w_{gk}} = \overline{z}_{w_g} + m_k \left( s_{w_g} + s_{w_0} \right) d_{w_{gk}}. \tag{17}$$

Next, each $d_{w_{gk}}$ is shrunken towards zero using the soft-thresholding shrinkage method. This method is equivalent to lasso. Using soft-thresholding with a certain threshold parameter $\lambda$, weighted shrunken differences can be obtained as follows:

$$d'_{w_{gk}} = sign \left( d_{w_{gk}} \right) max \left( \left| d_{w_{gk}} \right| - \lambda, 0 \right). \tag{18}$$

After shrinking $d_{w_{gk}} \rightarrow d'_{w_{gk}}$, we update the weighted centroids as follows:

$$\overline{z}'_{w_{gk}} = \overline{z}_{w_g} + m_k \left( s_{w_g} + s_{w_0} \right) d'_{w_{gk}}. \tag{19}$$

Increasing $\lambda$ will lead to obtaining sparser models by eliminating most of the genes from the class prediction. When $d'_{w_{gk}}$ is zero for a particular gene $g$, among all classes, the weighted centroids will be same across the classes. Hence, this gene will not contribute to the class prediction.

### Selection of the optimal threshold parameter ($\lambda$)

Selection of $\lambda$ is very important on the model sparsity. Increasing $\lambda$ will lead to obtaining sparser models. However, the predictive performance of obtained models might be decreased dramatically. Small values of $\lambda$, on the other hand, might increase the accuracy of classifiers, yet it may increase the complexity of the models. Thus, it is necessary to select $\lambda$ that yields both accurate and sparse results. Figure 2 displays the test set errors for a set of $\lambda$ parameters for the cervical dataset (*Witten et al., 2010*). It is clear that we obtain the minimum misclassification errors for the values of $\lambda = \{0.561, 0.654, 0.748, 1.028, 1.121, 1.215, 1.308, 1.402, 1.495, 1.588, 1.682, 1.775\}$. Among these values, selecting the maximum one will give us the sparsest solution. For this reason, we select the threshold to be 1.775 and obtain 96.5% accuracy by using only

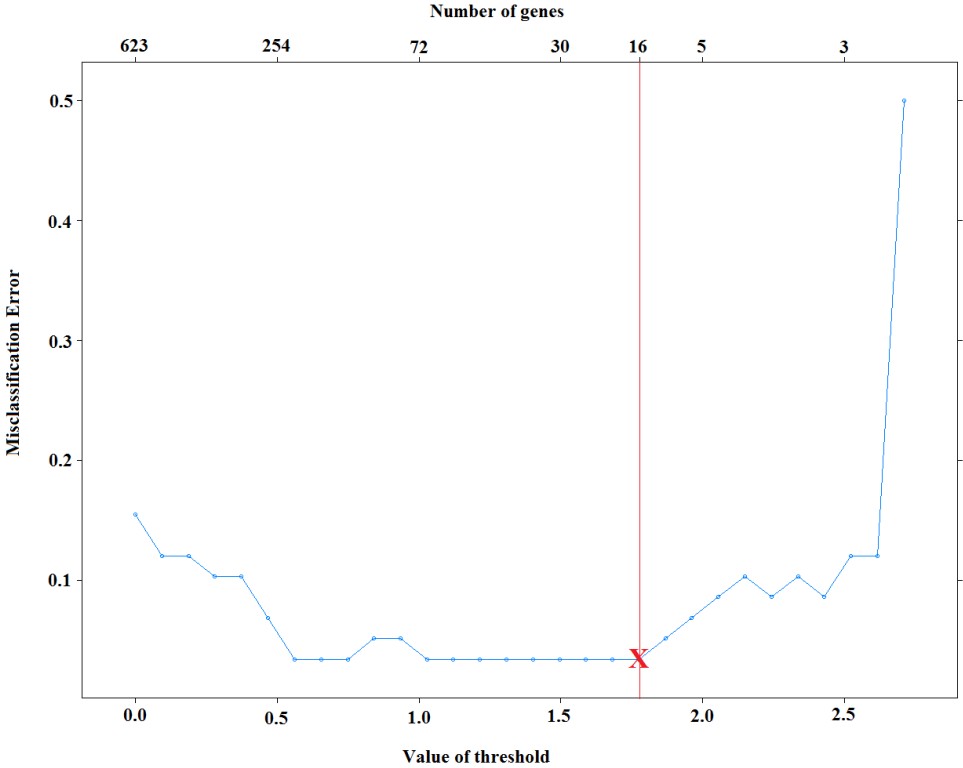

**Figure 2** Selection of voomNSC threshold parameter for cervical data.

16 features. Moreover, one can also use cross-validation technique and select the sparsest model that minimizes cross-validation error.

## Prediction of test observations for voomNSC classifier

Test observations are normalized and transformed based on the training set parameters, which was already explained in the previous sections. Again, a standardization is applied to the $z_{g*}$, log-cpm values of test observations, by the training parameters $s_{w_g} + s_{w_0}$. We classify a test observation to the class which maximizes the following discriminating function:

$$\delta_k^{voomNSC}(x_*) = -\frac{1}{2}\sum_{g=1}^{p}\frac{\left(z_{g*} - \overline{z}_{w_{gk}}\right)^2}{\left(s_{w_g} + s_{w_0}\right)^2} + \log(\hat{\pi}_k). \tag{20}$$

Finally, posterior probabilities can be obtained:

$$\hat{p}_k(x_*) = \frac{e^{-\delta_k(x_*)}}{\sum_{l=1}^{K} e^{-\delta_l(x_*)}}. \tag{21}$$

## Implementation of classifiers

To assess the performance of developed algorithms, we compared our results with several classifiers. In this section, we cover selected classifiers in detail.

Firstly, we selected discrete RNA-Seq classifiers (i.e., PLDA and NBLDA) for comparison, since they are the only algorithms proposed for RNA-Seq classification. We also applied

the diagonal discriminant classifiers (i.e., DLDA and DQDA), after transforming the data to make it more similar to microarrays. SVM (support vector machines) and RF (random forests) algorithms are also considered due to their good performances in microarray based classification studies. Implementation details of each algorithm, including voomDDA classifiers are given in below:

$PLDA_1$: The data are normalized using the DESeq median ratio method. Normalized count values are taken as input to the PLDA algorithm. A five-fold cross validation is performed to identify the optimal $\rho$ tuning parameter. A grid search (number of searches: 30) is applied and the sparsest model with the minimum misclassification error is selected to optimize $\rho$. The PLDA is applied with the optimum $\rho$ using PoiClaClu R package (*Witten, 2013*).

$PLDA_2$: After normalization, a power transformation ($X'_{ij} = \sqrt{X_{ij} + 3/8}$) is applied to reduce the effect of overdispersion and make genes have nearly constant variance. Normalized and transformed expression values are used as the input data for the PLDA algorithm. Other procedures are the same as with $PLDA_1$.

*NBLDA:* DESeq median ratio method is used for normalization. A shrinkage method is applied for estimation of the dispersion parameter as suggested by *Yu, Huber & Vitek (2013)*. The normalized count data are used as input for NBLDA algorithm. This algorithm is applied in R software with the necessary codes available in (*Dong et al., 2015*).

*NSC:* The DESeq median ratio method is used for the data normalization and the rlog transformation is applied to the normalized count data. The normalized and the transformed expression values are used as the input data for NSC algorithm. The proportions of class sample sizes are used as the class prior probabilities. A five-fold cross validation is used to determine the optimal threshold value. The optimum threshold value is obtained from the sparsest model with the minimum misclassification error after a grid search (number of searches: 30). NSC is applied using the R package pamr (*Hastie et al., 2014*).

*DLDA:* The DESeq median ratio method is applied for the data normalization and the rlog transformation is applied to the normalized count data. The normalized and the transformed expression values are used as the input data for DLDA algorithm. The proportions of class sample sizes are used as the class prior probabilities. Then the DLDA is applied using the R package sfsmisc (*Maechler, 2015*).

*DQDA:* Same procedure is applied with DLDA algorithm (*Maechler, 2015*).

*SVM:* The DESeq median ratio method is used for the data normalization and the rlog transformation is applied to the normalized count data. The normalized and the transformed expression values are used as the input data for SVM algorithm. A five-fold cross validation is performed and repeated three times, and a grid search (with tune length of 10) is made to determine the optimal sigma and cost parameters. The radial basis function is used to allow for nonlinear decision boundaries in the SVM. SVM is applied using the R package caret (*Kuhn, 2008*).

*RF*: The applied procedure is similar to SVM. Here, the optimized parameter is the number of variables randomly sampled as candidates at each split. Number of trees are set as 500. RF is applied using the R package caret (*Kuhn, 2008*).

*voomNSC*$_1$: The DESeq median ratio normalization is applied to the data and the normalized data is used as the input for the voomNSC classifier. The proportions of class sample sizes are used as the class prior probabilities. To optimize the threshold value, the sparsest model with the minimum misclassification error is selected. A grid search (number of searches: 30) is applied to determine the optimal threshold value.

*voomNSC*$_2$: The raw read counts are directly used as the input for the voomNSC algorithm. All other procedures remain same with voomNSC$_1$.

*voomNSC*$_3$: The TMM method is applied to normalize the data. The normalized data is used as the input for the voomNSC classifier. Other procedures are same with voomNSC$_1$ and voomNSC$_2$.

*voomDLDA*$_1$: The DESeq median ratio normalization is applied to the data and the normalized data is used as the input for voomDLDA classifier. The proportions of class sample sizes are used as the class prior probabilities.

*voomDLDA*$_2$: The raw count data are not normalized and directly used as the input for voomDLDA classifier. Other procedures are same with voomDLDA$_1$.

*voomDLDA*$_3$: The TMM method is used for normalization. Other procedures are same with voomDLDA$_1$ and voomDLDA$_2$.

*voomDQDA*$_1$: The DESeq median ratio normalization is applied to the data and the normalized data is used as the input for voomDQDA classifier. The proportions of class sample sizes are used as the class prior probabilities.

*voomDQDA*$_2$: The raw count data are not normalized and directly used as the input for voomDQDA classifier. Other procedures are same with voomDQDA$_1$.

*voomDQDA*$_3$: The TMM method is performed for normalization. Other procedures are same with voomDQDA$_1$ and voomDQDA$_2$.

## Evaluation of voomDDA classifiers

To evaluate the performance of the developed algorithms, we performed a comprehensive simulation study. Four real datasets were also used to illustrate the applicability of voomDDA classifiers and assess their performance in real experiments. All experimental R code is available in File S1.

## Simulation study

We simulated data ($p \times n$ dimensional matrix) under 648 scenarios using negative binomial distribution as follows:

$$x_{gi}|y_i = k \sim NB\left(\mu_{gi}e_{gk}, \phi_g\right) g = 1, \ldots, p; \ i = 1, \ldots, n; \ k = 1, \ldots, K \qquad (22)$$

where NB corresponds to negative binomial distribution, $\mu_{gi}$ corresponds to $g_g s_i$, $e_{gk}$ is the differential expression probability for each of the $p = 10,000$ genes among classes, and $\phi_g$ is the dispersion parameter. For a given $y_i = k$, $x_{gi}$ has mean $\mu_{gi}e_{gk}$ and variance $\mu_{gi}e_{gk} + \left(\mu_{gi}e_{gk}\right)^2 \phi_g$. $s_i$ is the size factor for each sample and simulated identically and independently from $s_i \sim Unif(0.2, 2.2)$. $g_g$ refers to the total number of counts per gene and also simulated identically and independently from $g_g \sim Exp(1/25)$. If a gene is not differentially expressed between classes $k$, then $e_{gk}$ is set to 1. Otherwise, $\log(e_{gk}) = \tilde{z}_{gk}$,

where the $\tilde{z}_{gk}$'s are identically and independently distributed from $\tilde{z}_{gk} \sim N\left(0, \sigma^2\right)$. $\sigma$ is set to 0.10 or 0.20 in simulations. Of the total $p = 10{,}000$ genes, 500, 1,000 and 2,000 genes with maximum variances are selected. We added a small constant ($\varepsilon = 1$) to count values of each simulated data to avoid taking the logs of zero in the following analysis.

The simulated datasets contain all possible combinations of:

- number of genes; $p' = (500, 1{,}000, 2{,}000)$,
- number of biological samples; $n = (40, 60, 80, 100)$,
- number of classes; $K = (2, 3, 4)$,
- probability of differential expression: $e_{gk} = (1\%, 5\%, 10\%)$,
- standard deviation parameter: $\sigma = (0.1, 0.2)$
- dispersion parameter; ($\phi_g = 0.01$: very slight, $\phi_g = 0.1$: substantial; $\phi_g = 1$, very high overdispersion).

Simulation code for generating count data from NB distribution are adapted from the *CountDataSet* function of the PoiClaClu R package (*Witten, 2013*) based on the simulation details given above. Seed number is set to a constant of '10072013' for random numbers generation.

The following steps are applied in the exact order after count data are simulated. A flow chart is provided for the reader to better understand the evaluation processes (File S1).

*Data splitting:* The data are randomly split into training and test sets with 70% and 30% of the data, respectively. The feature data can be denoted as $X_{\text{tr}}$ and $X_{\text{ts}}$, where the class labels can be denoted as $y_{\text{tr}}$ and $y_{\text{ts}}$.

*Near-zero filtering:* Since the genes with low counts can affect the further analysis (e.g., linear modeling inside voom transformation), genes having near zero variances in the training set are filtered in this step. For this purpose, two criteria are used for filtering: (i) the ratio of the most frequent value to the most frequent second value is higher than 19 (95/5); (ii) the number of unique values divided by the sample size is less than 10%. The genes with near zero variance are filtered from the test set as well.

*Variance filtering:* Next, a second filtering is applied to keep only the informative genes in the model. In the training set, 500, 1,000 and 2,000 genes with maximum variances are selected and other genes are filtered from both training and test sets. In this step, count data are normalized using the DESeq median ratio method and transformed using vst transformation. The genes are sorted in decreasing order based on their variances. The count values of the selected genes were fetched again for further analysis.

*Normalization:* After filtering steps, the datasets are normalized to adjust the sample specific differences using the DESeq median ratio method or TMM method depending on the selected classification method. Note that the size factors required for the normalization are calculated from the unfiltered raw dataset. The datasets are not normalized for voomNSC$_3$, voomDLDA$_3$ and voomDQDA$_3$ classifiers. Since training and test sets should be in the same scale and be homoscedastic relative to each other, the normalization of test datasets is made based on the information obtained from the training datasets. Therefore, each test sample should be independently normalized using the same parameters calculated from the training set as described in the previous section.

*Transformation:* After normalization, several transformations are applied to the data to estimate the mean and the variance relationship of the data. Normalized count data are converted into a continuous scale using this mean and variance relationship. Since PLDA$_1$ and NBLDA are count-based classifiers, the transformations are not applied for these classifiers. VoomDDA classifiers use the voom method inside the algorithm for transformation. A power transformation is applied for PLDA$_2$ classifier. The rlog transformation is performed for other classifiers, due to its capability of accounting for variations in sequencing depth across samples (*Love, Huber & Anders, 2015*). Similar to the normalization, the test sets are transformed based on the mean and variance relationship (of genes or samples) of the training sets. Thus, we do not re-estimate the mean–variance relationship in the sets. The same $\beta_g$ coefficients are used for both training and test sets.

*Model fitting and parameter optimization:* In order to avoid overfitting and underfitting, we optimized the tuning parameters of classifiers before model fitting. A five-fold cross validation is performed on the training set and the parameter that gives the minimum misclassification error is identified as optimal. Same folds are used in all classifiers to make the results comparable. In case of equal misclassification errors, the best parameter is chosen based on its sparsity. Finally, classification models are fitted on $X_{tr}$ and $y_{tr}$ with the optimal tuning parameters.

*Prediction and performance evaluation:* The optimal model is obtained from training data and new test samples are classified into one of the possible classes. The misclassification error is calculated for each classifier. The number of genes used in each model is also saved in order to assess sparsity.

Since we mimic the real datasets, sample sizes are set to be very small relative to the number of genes. Thus, the misclassification errors may be highly variable depending on the split of samples into training and test sets. To overcome this problem, all the entire simulation procedure was repeated 50 times and the summaries are given in the Results section.

## Application to real RNA-sequencing datasets
### Experimental datasets
*Cervical dataset:* The cervical dataset is a miRNA sequencing dataset obtained from *Witten et al. (2010)*. The objective of this study was to both identify the novel miRNAs and to detect the differentially expressed ones between normal and tumor cervical tissue samples. For this purpose, the authors constructed 58 small RNA libraries, prepared from 29 cervical cancer and 29 matched control tissues. After deep sequencing with Solexa/Illumina sequencing platform, they obtained a total of 25 Mb and 17 Mb RNA sequences from the normal and the cancer libraries respectively. Of these 29 tumor samples, 21 of them had a diagnosis of squamous cell carcinoma, six of them had adenocarcinoma and two were unclassified. In our analysis, we used the data that contains the sequence read counts of 714 miRNAs belonging to 58 human cervical tissue samples, where 29 tumor and 29 non-tumor samples are treated as two distinct classes for prediction.

*Alzheimer dataset:* This dataset is another miRNA dataset provided from *Leidinger et al. (2013)*. The authors aimed to discover potential miRNAs from blood in diagnosing

Alzheimer and related neurological diseases. For this purpose, the authors obtained blood samples from 48 Alzheimer patients that were evaluated after undergoing some tests, including Alzheimer Disease Assessment Scale-cognitive subscale (ADAS-Cog), Wechsler Memory Scale (WMS), and Mini-Mental State Exam (MMSE) and Clinical Dementia Rating (CDR). A total of 22 age-matched control samples were obtained and all sample libraries were sequenced using Illumina HiSeq2000 platform. After obtaining the raw read counts, the authors filtered the miRNAs with less than 50 counts in each group. We used the data, including 416 read counts of 70 samples, where 48 Alzheimer and 22 control samples are considered as two separate classes for classification.

*Renal cell cancer dataset:* Renal cell cancer (RCC) dataset is an RNA-Seq dataset that is obtained from The Cancer Genome Atlas (TCGA) (*Saleem et al., 2013*). TCGA is a comprehensive community resource platform for researchers to explore, download, and analyze datasets. We obtained the raw 20,531 known human RNA transcript counts belonging to 1,020 RCC samples from this database (with options level 3, RNASeqV2 data). This RNA-Seq data has 606, 323 and 91 specimens from the kidney renal papillary cell (KIRP), kidney renal clear cell (KIRC) and kidney chromophobe carcinomas (KICH), respectively. These three classes are referred as the most common subtypes of RCC (account for nearly 90%–95% of the total malignant kidney tumors in adults) and treated as three separate classes in our analysis (*Goyal et al., 2013*).

*Lung cancer dataset:* Lung cancer is another RNA-Seq dataset provided from TCGA platform. Same options as RCC data were used in the download process. The resulting count file contains the read counts of 20,531 transcripts of 1,128 samples. The dataset has two distinct classes including lung adenocarcinoma (LUAD) and lung squamous cell with carcinoma (LUSC) with 576 and 552 class sizes, respectively. These two classes are used as class labels in our analysis.

Batch effects of TCGA data sets including renal cell cancer and lung cancer types were removed using the Combat function in the SVAseq package (*Leek, 2014*) prior to classification.

## Evaluation process

A similar procedure is followed with the simulation study. The data are randomly split into two parts as training (70%) and test (30%) sets. Near zero filtering is applied to all datasets except Alzheimer, since low counts were already filtered by the authors of the study (*Leidinger et al., 2013*). Next, 2,000 transcripts with the highest variances are selected in each of the renal cell cancer and lung datasets. Appropriate normalization, transformation and model fitting processes are applied same with the simulation study. In prediction step misclassification errors for Alzheimer and renal cell cancer datasets are balanced due to the unbalanced class sizes.

We repeated the entire process 50 times, since cervical and Alzheimer datasets have relatively small sample size. The test set errors may differ for different train/test splits. Seed number is set between 1 to 50 in the analysis steps. In the results section, summary statistics are given across these 50 repeats.

## Evaluation criteria

To assess the performance of classifiers, we used three criteria: (i) sparsity, (ii) accuracy and (iii) computational cost. We simply assessed the sparsity of each model by calculating the sparsity which is the number of selected genes in each model, or relative sparsity, with the ratio of the number of genes selected in each classification model over a total number of genes. A model with the lower number of genes in the decision rule is considered as the sparser model. We calculated misclassification errors as the accuracy measure of each model in the test set. Due to the high-dimensionality of the RNA-Seq data, it is possible to encounter the overfitting problem. This means a classification method may perfectly classify the training set, but may not perform well in the test set. Since correctly predicting the class labels of new observations is the major purpose in real life problems, we randomly split each dataset into training and test sets. All model building processes are applied in the training set and the performance assessment is performed on test sets. Misclassification errors are calculated from predicted test observations. A model with less misclassification error is considered as the more accurate model.

In case of unbalanced class sizes, misclassification error may lead to problems measuring the actual accuracy. Here, we used the balanced misclassification error as evaluation criterium: $(1 - (Sensitivity + Specificity)/2)$. For multiclass problems, performance metrics are calculated by the one-versus-all method and by comparing each class label to the remaining labels.

# RESULTS

## Simulation results

Misclassification errors and sparsity results for the simulation scenario $K = 2$, $e_{gk} = 5\%$, $\sigma = 0.1$ are given in Figs. 3 and 4. Entire simulation results for each scenario are given in File S2. These figures differ with different combinations of the number of classes ($K$), the probability of differential expression ($e_{gk}$) and standard deviation ($\sigma$). Odd numbered figures give the accuracy results, while the even numbered figures give the sparsity results. Note that the sparsity results are only given for sparse classifiers (i.e., NSC, PLDA$_1$, PLDA$_2$, voomNSC$_1$, voomNSC$_2$, voomNSC$_3$). All figures are given in the same format in the same matrix layout. Each figure displays the effect of sample size ($n$), the number of genes ($p'$), dispersion parameter ($\phi_g$) on the accuracy and the sparsity of classification models. Axis panels give the results for sample size, ordinate panels give the results for dispersion parameter. Each panel demonstrates the error bars for each classifier on classification performance. Classifiers are displayed in axes; evaluation measures are displayed in ordinates in each panel. Each measure is in the range $[0,1]$, where lower values corresponding to more accurate or sparse models. Error bars are generated from the arithmetic mean and 95% confidence level of each performance measure in 50 repeats. Black, red and green bars correspond to the results for 500, 1,000 and 2,000 genes, respectively.

As can be seen from the figures, an increase in the sample size leads to an increase in the overall accuracies, unless the data are overdispersed. This relation is more distinct for

 

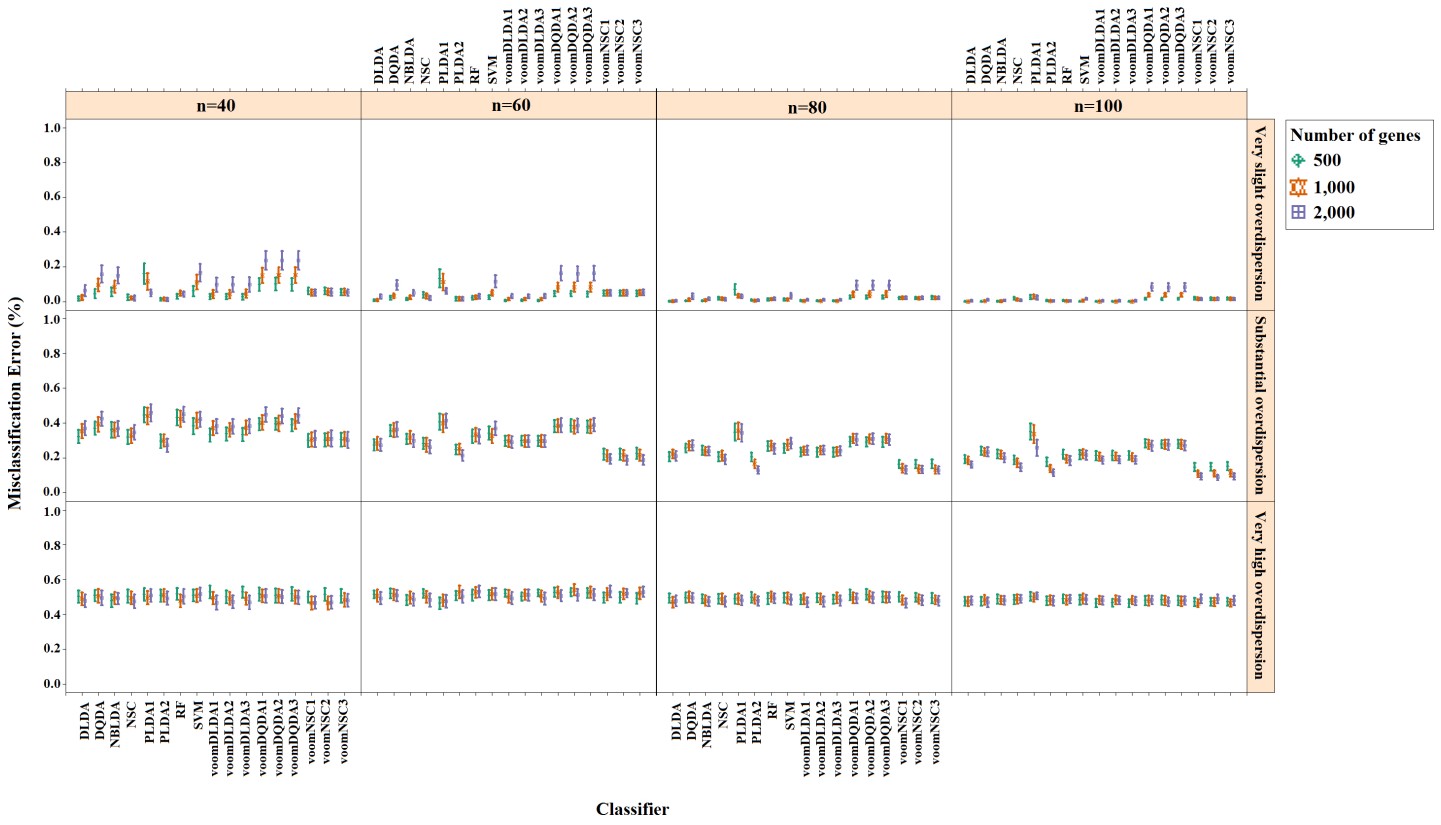

**Figure 3  Misclassification errors of classifiers for the simulation scenario $K = 2$, $e_{gk} = 5\%$, $\sigma = 0.1$.**

very slightly overdispersed scenarios. However, this increase does not affect the amount of sparsity. The number of genes has a considerable effect on both accuracy and sparsity. Including more genes into classification models, mostly leads to more accurate results for PLDA (PLDA$_1$, PLDA$_2$) and voomNSC (voomNSC$_1$, voomNSC$_2$, voomNSC$_3$) classifiers, unless the data are overdispersed. Increasing the number of genes, mostly provides less accurate results for other classifiers. However, this relation may change in some scenarios where the sample size and the standard deviation increased. VoomNSC and PLDA classifiers mostly produce sparser results depending on the increase in the number of genes. This situation is quite opposite for the NSC algorithm in most scenarios.

The change in dispersion parameter has a direct effect on both model accuracies and sparsities. When the data become more spread, the amount of accuracy decreases in all classifiers. In slightly overdispersed data, all classifiers, except NSC, produce sparser results. NSC gives sparser solutions based on the increase in the differential expression probability. The increase in this probability causes less sparse solutions for PLDA classifier in many scenarios.

Increasing the number of classes lead to a decrease in classification accuracies. This relation particularly becomes apparent when the number of genes and the differential expression probability increases as well. The decrease in the performance of PLDA$_2$ and voomNSC classifiers is less than the other algorithms. An increase in the class

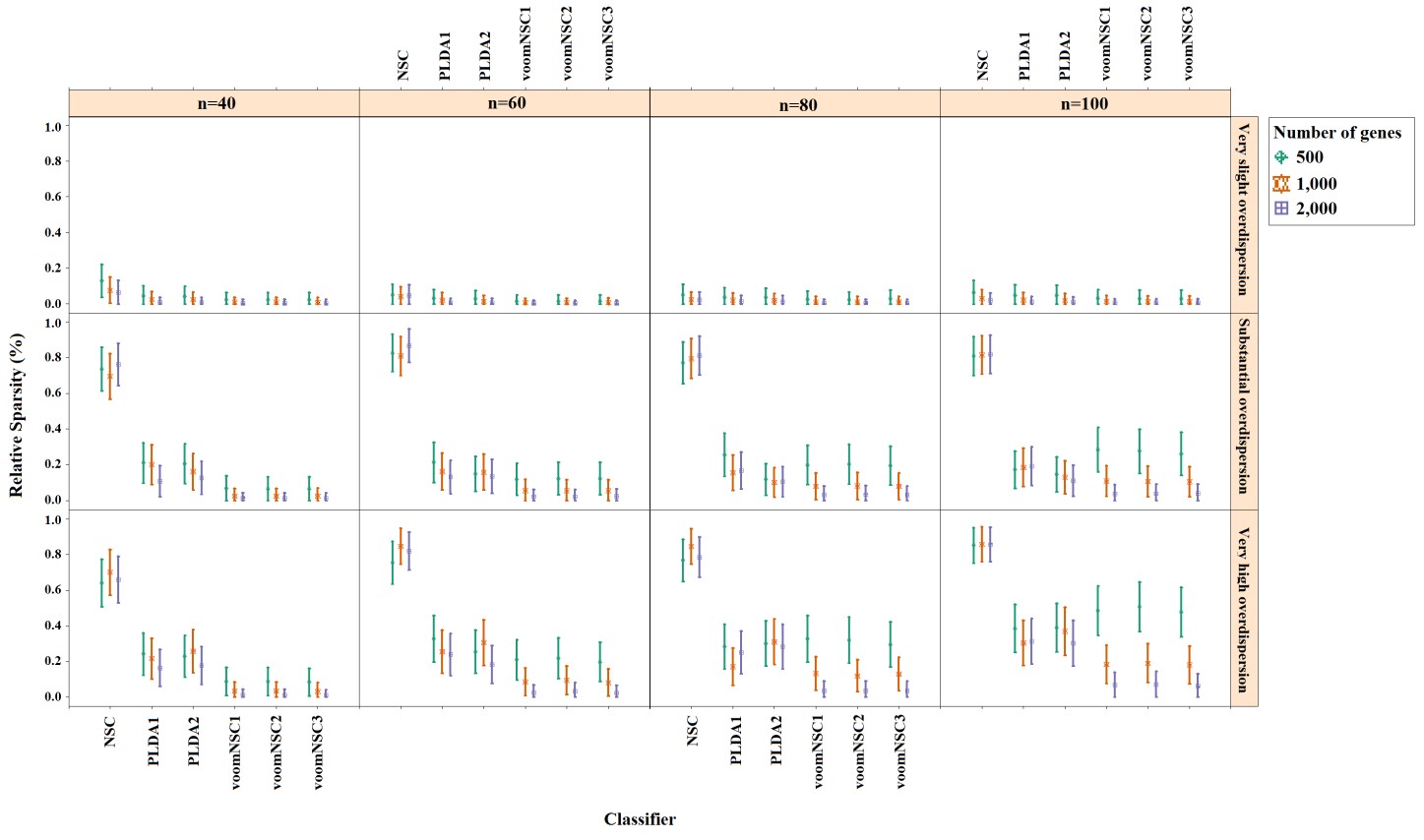

**Figure 4** Sparsities of classifiers for the simulation scenario $K = 2$, $e_{gk} = 5\%$, $\sigma = 0.1$.

numbers affects the sparsity of NSC classifier in a negative way, while does not affect the other classifiers.

Nearly all classifiers demonstrate higher accuracy if the differential expression probability increases. Only PLDA$_1$ and NSC classifiers show less accurate performances in this situation depending on the increase in the standard deviation. The increase in these differential expression probabilities brings sparser model performances, mostly for NSC algorithm in slightly overdispersed datasets. The increase in the standard deviation leads to more accurate and sparser classification algorithms. This may be different for the NSC classifier in slightly overdispersed datasets. This case results in sparser solutionsWhen we assess the performances of classifiers relative to each other, PLDA$_2$ and voomNSC classifiers are the most accurate algorithms for slightly overdispersed datasets. PLDA$_1$ may be considered as the second best performer, RF as the third performer and NSC as the fourth performer among all classifiers. In substantially overdispersed datasets, voomNSC classifiers are the most accurate classifiers, mostly for the scenarios with a high number of genes. PLDA$_2$ gives compatible results with these classifiers. RF provides substantial results behind voomNSC and PLDA$_2$ classifiers. In highly overdispersed data, all methods generally give very poor results. Considerable performances may be seen when the number of class decreases, and the number of samples, differential expression probability, and the standard deviation

increases. In such cases, again $PLDA_2$ and voomNSC classifiers outperform other classifiers, mostly for the scenarios with a large number of genes.

In slightly overdispersed datasets, all methods, except for the NSC algorithm, provide very sparse results. Sparser results for NSC algorithm are obtained with the increase in probability in differential expression and standard deviation. In datasets with substantial overdispersion, voomNSC classifiers seem to show their ability, and produce sparser models than the other classifiers, especially in scenarios with high number of genes. In highly overdispersed datasets, voomNSC classifiers clearly build the sparsest models. In this case, PLDA classifiers give less sparse solutions, while the NSC algorithm gives the poorest results.

Nonsparse voomDDA classifiers gave compatible results with the rlog generalizations of DLDA and DQDA classifiers. Dispersion has a significant effect on PLDA classifier and $PLDA_2$ classifier outperforms $PLDA_1$ in both accuracy and sparsity in most scenario.

As an overall evaluation of the classifiers, we can say that $PLDA_2$ and voomNSC classifiers outperform other classifiers based on the accuracies. When we consider the sparsity measure, voomNSC classifiers are the overall winner and provide the sparsest solutions compared to the other methods. Finally, we note that the normalization does not have a significant effect on the performance of the voomNSC algorithm, since all three forms of this method performed very similarly.

### Results for real datasets

Results for real datasets are given in Tables 1 and 2. Misclassification errors are given in Table 1 and the amount of sparsities is given Table 2 for each classifier across 50 repetitions.

In cervical dataset, NBLDA, SVM and NSC algorithms gave the most accurate results with 8.9%, 10.1% and 10.8% misclassification errors, respectively. NBLDA and SVM algorithms use all miRNAs for prediction while NSC selected an average of 194 from all features. The error rates for voomNSC and $PLDA_2$ classifiers were between 11–12%. An average of 290 miRNAs was selected for $PLDA_2$ classifier, while this number was between 56.28 and 63.34 for voomNSC classifiers. Thus, voomNSC classifiers can be considered as the best performers, since the average test errors were compatible with NBLDA, SVM and NSC algorithms; however, they use substantially fewer miRNAs than the other classifiers.

In Alzheimer dataset, SVM and $voomDQDA_2$ algorithms performed more accurately than the other algorithms with 8.7% and 13.9% misclassification errors, respectively. $PLDA_1$ was the sparsest classifier with an average of 11 miRNAs. However, its test error was 31.7%, which is much higher than for the other algorithms. Among the other sparse classifiers, $voomNSC_3$ and $voomNSC_1$ fit the model with an average of 30 and 48 miRNAs, respectively. Thus, SVM and $voomNSC_3$ classifiers can be considered as the best performers. For this dataset, SVM builds more accurate but also more complex models. On the other hand, $voomNSC_3$ classifier gives sparser, but less accurate results than the SVM algorithm.

In renal cell cancer dataset, SVM and RF are the most accurate classifiers with 6.5% and 7.7% misclassification errors, respectively. The $PLDA_1$ classifier was the poorest algorithm with 75.6% test set error. The performance of the voomNSC classifier was around 18–19%, which is less accurate than other algorithms. Misclassification error rates for other classifiers

**Table 1 Misclassification errors of classifiers for real datasets.**

| Classifier | Cervical | Alzheimer | Renal cell cancer | Lung Cancer |
|---|---|---|---|---|
| DLDA | 0.149(0.015) | 0.197(0.012) | 0.140(0.003) | 0.098(0.002) |
| DQDA | 0.140(0.012) | 0.188(0.012) | 0.135(0.003) | 0.098(0.002) |
| NBLDA | **0.089(0.010)** | 0.198(0.014) | 0.139(0.003) | 0.098(0.002) |
| NSC | 0.108(0.011) | 0.201(0.012) | 0.140(0.003) | 0.097(0.002) |
| $PLDA_1$ | 0.287(0.029) | 0.317(0.014) | 0.756(0.044) | 0.262(0.028) |
| $PLDA_2$ | 0.111(0.011) | 0.223(0.013) | 0.143(0.003) | 0.100(0.002) |
| RF | 0.135(0.012) | 0.204(0.013) | 0.077(0.002) | 0.062(0.002) |
| SVM | 0.101(0.010) | **0.087(0.010)** | **0.065(0.002)** | **0.052(0.002)** |
| $voomDLDA_1$ | 0.148(0.015) | 0.210(0.012) | 0.141(0.003) | 0.097(0.002) |
| $voomDLDA_2$ | 0.211(0.019) | 0.228(0.015) | 0.139(0.003) | 0.097(0.002) |
| $voomDLDA_3$ | 0.146(0.015) | 0.203(0.012) | 0.142(0.003) | 0.097(0.002) |
| $voomDQDA_1$ | 0.164(0.014) | 0.181(0.012) | 0.134(0.002) | 0.097(0.002) |
| $voomDQDA_2$ | 0.165(0.013) | 0.139(0.010) | 0.138(0.003) | 0.098(0.002) |
| $voomDQDA_3$ | 0.153(0.014) | 0.170(0.011) | 0.137(0.003) | 0.095(0.002) |
| $voomNSC_1$ | 0.119(0.013) | 0.227(0.010) | 0.181(0.002) | 0.097(0.002) |
| $voomNSC_2$ | 0.111(0.010) | 0.226(0.018) | 0.192(0.003) | 0.097(0.002) |
| $voomNSC_3$ | 0.112(0.012) | 0.233(0.012) | 0.184(0.002) | 0.092(0.002) |

**Notes.**
Values are misclassification errors, calculated from 50 repetitions and expressed as mean (standard error). Best performed methods are indicated as bold in each column.

**Table 2 Sparsities of classifiers for real datasets.**

| Classifier | Cervical | Alzheimer | Renal cell cancer | Lung cancer |
|---|---|---|---|---|
| NSC | 194.18(27.40) | 333.06(19.04) | 1989.00(7.32) | 1685.22(47.73) |
| $PLDA_1$ | 290.44(40.01) | **10.81(9.31)** | 606.82(112.40) | 1339.90(112.54) |
| $PLDA_2$ | 126.66(29.13) | 228.97(22.53) | 1640.47(81.59) | 1060.84(70.93) |
| $voomNSC_1$ | **56.28(10.94)** | 48.06(10.78) | **178.26(8.18)** | 85.04(39.34) |
| $voomNSC_2$ | 59.16(13.60) | 140.32(20.22) | 700.90(114.63) | 122.44(33.22) |
| $voomNSC_3$ | 63.34(13.94) | 30.02(8.10) | 208.22(42.35) | **54.18(34.97)** |

**Notes.**
Values are the number of genes selected in each model, calculated from 50 repetitions and expressed as mean (standard error). Best performed methods are indicated as bold in each column.

were between 13–15%. When we look at the sparsity results, NSC and $PLDA_2$ classifiers provided less sparse solutions, with an average of 1,989 and 1,649 genes, respectively. $PLDA_1$ and $voomNSC_2$ obtained moderate sparsity results with an average of 607 and 701 genes, respectively. On the other hand, $voomNSC_1$ and $voomNSC_3$ gave the sparsest results for this dataset. In this dataset, $VoomNSC_1$ selected an average of 178 genes, while $voomNSC_3$ selected 202 genes. In the light of these results, we recommend using SVM and RF classifiers to obtain more accurate results and recommend $voomNSC_1$ and $voomNSC_3$ for sparsest results.

In lung cancer dataset, SVM and RF methods are again the most accurate classifiers with 5.2–6.2% test set errors, respectively. $PLDA_1$ performed as the less accurate algorithm with a 26.2% misclassification error. The performance of other classifiers was quite similar

**Table 3   Summary of voomNSC models and selected genes in real datasets.**

| Classifier | Misclassification error | Number of features | Selected features |
|---|---|---|---|
| Cervical | 2/58 | 14 | miR-1, miR-10b*, miR-147b, miR-183*, miR-200a*, miR-204, miR-205, miR-21*, miR-31*, miR-497*, miR-542-5p, miR-944, Candidate-5, Candidate-12-3p |
| Alzheimer | 13/70 | 3 | miR-367, miR-756, miR-1786 |
| Renal cell cancer | 87/1,020 | 87 | SLC6A3, RHCG, CA9, ATP6V0A4, CLDN8, TMEM213, FOXI1, SLC4A1, PVALB, KLK1, DMRT2, ATP6V0D2, PTGER3, HEPACAM2, CLCNKB, BSND, LCN2, PLA2G4F, SLC17A3, ATP6V1G3, RHBG, SLC9A4, GCGR, CLCNKA, NR0B2, CFTR, SCEL, ATP6V1B1, NDUFA4L2, FGF9, ENPP3, TMPRSS2, WBSCR17, HAPLN1, ACSM2A, FLJ42875, C6orf223, SLC26A7, ACSM2B, LRP2, FBN3, CNTN6, UGT2A3, EPN3, CALCA, SLC22A11, KLK4, STAP1, LOC389493, FOXI2, CLRN3, HS6ST3, HAVCR1, PART1, EBF2, PCSK6, SLC28A1, SFTPB, OXGR1, CLNK, C16orf89, HSD11B2, TRIM50, ACMSD, CXCL14, VWA5B1, KLK15, INPP5J, LRRTM1, SYT7, HGFAC, FAM184B, C1orf186, KLK3, GPRC6A KBTBD12, HCN2, C9orf84, GCOM1, PCDH17, PDZK1IP1, KRTAP5-8, ODAM, RGS5, CTNNA2, GGT1, KDR |
| Lung cancer | 96/1,118 | 6 | DSG3, CALML3, KRT5 , SERPINB13, DSC3, LASS3 |

and lies between 9.2% and 10.0%. NSC and PLDA classifiers gave substantially less sparse solutions than voomNSC classifiers. The number of selected genes was approximately 1,685 genes for NSC, 1,340 and 1,061 genes for $PLDA_1$ and $PLDA_2$, between 54 and 122 genes for voomNSC classifiers.

A summary for the selected genes in each real dataset is given in Table 3.

## Computational cost of classifiers

Along with the accuracy and sparsity results, we calculated the computational costs of each classifier to see whether the developed algorithms are applicable to real datasets. We used a workstation with the properties of Xeon E5-1650, 3.20 GHz CPU, 64GB memory and 12 cores. Performance results are given in File S2. All classifiers seem to be practical for cervical and Alzheimer miRNA datasets. These classifiers are able to fit models less than 2.15 s, for both datasets. Both sample size and number of features are higher in the renal cell and lung cancer datasets relative to the other data. This increase affects the computational performance of classifiers, mostly for RF and SVM. In general, DLDA and DQDA classifiers are the fastest among these classifiers. Moreover, the computational performance of voomDDA classifiers is also considerable, which is between 0.16 and 5.07 s in all datasets.

### voomDDA web-tool

To provide the applicability of the developed approaches, a user-friendly web application is developed with the shiny package of R. This tool is an interactive platform, which can be accessed through http://www.biosoft.hacettepe.edu.tr/voomDDA/. All source codes are available on GitHub (https://github.com/gokmenzararsiz/voomDDA) and in File S3. The tool includes the sparse voomNSC, non-sparse voomDLDA and voomDQDA algorithms accompanied with several interactive plots.

Users can upload either miRNA or mRNA based gene-expression data to identify the diagnostic biomarkers and to predict the classes of test cases. Two example datasets are also
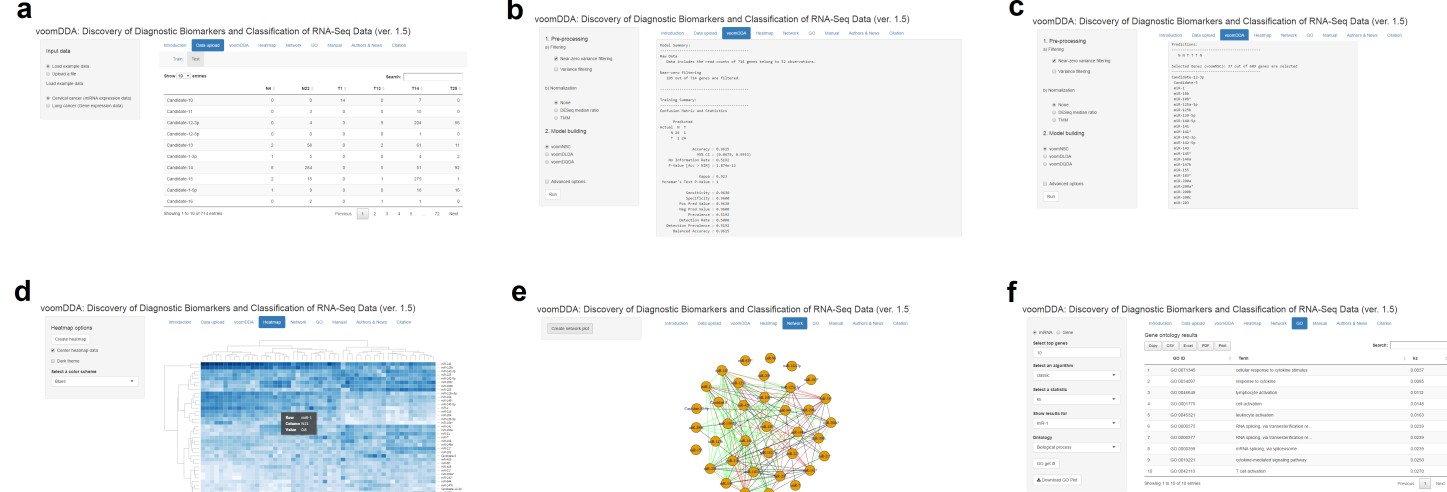

**Figure 5** Illustration of voomDDA web-tool.

available in the web-tool for the implementation of the developed approaches (Fig. 5A). After uploading the data, users can preprocess their data (i.e., filter and normalize), and build a classification model either using voomNSC, voomDLDA or voomDQDA. After selecting any of the three classifiers, a summary of the fitting process is displayed on the screen. A confusion matrix and several statistical diagnostic measures are given to examine the performances of classifiers (Fig. 5B). Based on the trained model, predictions and the identified markers will be displayed in the main panel of the tool, if the test set is uploaded by the user. Otherwise, voomDDA only displays the identified markers from the sparse voomNSC classifier (Fig. 5C). Furthermore, users can carry out various downstream analyses, i.e., heatmap plots, gene-network analysis and GO analysis, with the identified markers (Fig. 5D, Fig. 5E, and Fig. 5F). More detailed information can be found on the manual page of the web-tool.

## DISCUSSION

Biomarker discovery and classification are crucial in medicine to assist physicians and other health professionals in decision-making tasks, such as determining a diagnosis based on patient data. With the use of the capabilities of next-generation sequencing technology, detecting the most relevant genes (or exons, transcripts and isoforms) associated with a condition of interestand developing a decision support system for clinical diagnosis will enable physicians to make a more accurate diagnosis, develop and implement personalized, patient centered therapeutic interventions and improve the life quality of patients by better treatments.

In this study, we presented a sparse classifier voomNSC for classification of RNA- Seq data. We successfully coupled the voom method and the NSC classifier together by using weighted statistics, thus extended voom method for classification studies and made NSC algorithm available for RNA-Seq data. We also proposed two non- sparse classifiers, which

are the extensions of DLDA and DQDA algorithms for RNA-Seq classification. *Law et al. (2014)* introduced the voom method for differential expression analysis and for gene set testing. The authors stated that using precision weights with appropriate statistical algorithms may increase the predictive power of classifiers. We now extended this method to classification analysis and obtained very accurate and sparse algorithms.

We designed a comprehensive simulation study and also used four real miRNA and mRNA sequencing datasets to assess the performance of developed approaches and compare their performances with other classification algorithms. We obtained good results in both simulated and real datasets. In particular, voomNSC is able to find small subset of genes in an RNA-Seq data and provides fast, accurate and sparse solutions for RNA-Seq classification.

We compared our results with both the count based RNA-Seq classifiers and the microarray based classifiers after rlog transformation. To the best of our knowledge, count based classifiers are the only developed approaches in the literature for RNA-Seq data analysis. Using microarray based classifiers; we were able to see the effect of the voom method in classification studies. We selected the rlog transformation for microarray based classifiers, since it is good at accounting for the differences in library size. It also stabilizes the variances more accurately than a simple logarithmic transformation. In the simulation studies, the provided precision weights of voom method led to both more accurate and sparse models than obtained with microarray based classifiers. $PLDA_2$ gives compatible results with voomNSC classifiers in terms of classification accuracy. However, voomNSC provides sparser models, which is crucial for simpler, interpretable and low variance models. In real datasets, the accuracy results of the classifiers were comparable with each other. However, again the voomNSC classifiers provided the sparsest solutions.

Our approaches are mostly superior to PLDA, NBLDA, DLDA, DQDA, and NSC in providing sparser and models with comparable accuracy. $PLDA_2$ and voomNSC classifiers give comparable results in model accuracy. We believe that this superiority originates from the robustness of the voom methodology. This method empirically estimates the mean–variance relationships from the datawhile both PLDA and NBLDA aim to specify the exact probability distribution of counts instead. Precision weights allow us to make use of the normal distribution, since its mathematical theory is more tractable than the count distributions (*Law et al., 2014*). Precision weights also provide advantages such as working with samples with different sequencing depths, or the possibility of down-weighting the low-quality samples.

Dispersion has a direct effect on the PLDA classifier. The reason may be that the PLDA algorithm uses a Poisson model which assumes that the mean and the variance are equal. Nevertheless, applying a power transformation enhances its performance. Thus, we recommend users to always use the PLDA classifier with power transformation, since RNA-Seq data is mostly overdispersed, because of the presence of biological replicates in most datasets. Overdispersion has a significant effect on this classifier and should be taken into account before building models. The NBLDA classifier (*Dong et al., 2015*) converges to the PLDA algorithm, when the dispersion parameter approximates to zero. This classifier performed well for the overdispersed cervical datasets; however, it does not perform as well

as the PLDA$_2$ classifier or the voomNSC classifier in other scenarios. This may originate from the absence of a sparsity option in this classifier. We leave sparse NBLDA classification as a topic for further research.

In slightly overdispersed datasets, RF performs as well as the sparse classifiers. Moreover, this classifier performed very well in lung and renal cell cancer datasets. One possible explanation for this result may be the bootstrap property of this algorithm. As in microarray classification, the SVM algorithm performed very accurately in real datasets. *Mukherjee et al. (1999)* stated that this high accuracy may arise because of the strong mathematical background of SVM algorithm. The idea here is that the margin overcomes the overfitting problem and make SVM algorithm capable to work in high-dimensional settings. This is also true for RNA-Seq datasets, since rlog transformation makes the RNA-Seq more similar to microarray data.

When we increased the number of classes, the overall accuracy was decreased. This may arise because of the decrease of assignment probability of a sample in this condition. Moreover, we saw that the effect of sample size and number of genes on misclassification errors is highly dependent on the dispersion parameter. Decreasing the number of genes and samples leads to an increase in the misclassification error, unless the data are overdispersed.

Normalization had little impact on voomDDA classifiers in simulation results. However, we observed that performing voomNSC algorithm without any normalization provides less sparse results in Alzheimer, lung and renal cell cancer datasets. This may arise because of the very large differences in library sizes (e.g., 2.6 to 100.6 million in Alzheimer dataset). In this case, the DESeq median ratio or TMM methods can be applied before model building in order to obtain sparser results. In other cases, all three voomNSC classifiers provided very similar results in both model accuracy and sparsity. This result is consistent with *Witten (2011)*. Normalization may significantly affect the results in data with few features with very high counts.

We also demonstrated the use of the voomNSC algorithm in diagnostic biomarker discovery problems. In the cervical dataset, voomNSC identified 14 miRNAs as biomarkers with misclassifying two out of 58 samples. *Witten et al. (2010)* applied NSC algorithm in their paper and identified 41 miRNAs. A total of 9 miRNAs detected by the voomNSC algorithm, including miR-200a*, miR-204, miR-205, miR- 1, miR-147b, miR-31*, miR-944, miR-21*, and miR-10b* were identified to be common with the authors (Fig. 6). Moreover, voomNSC also used Candidate-5, Candidate-12-3p, miR-183*, miR-497*, and miR-542-5p in the prediction. *Witten et al. (2010)* misclassified four out of 58 samples. Thus, our algorithm is superior to their procedure in both accuracy and sparsity for classifying this dataset.

*Leidinger et al. (2013)* identified 12 miRNAs in classifying Alzheimer data and obtained 7% misclassification errors. In our study, we detected three miRNAs and obtained 18.6% misclassification error rate. Any of the selected miRNAs were found to be common with each other. Hence, voomNSC performed less accurate, however, sparser solutions than their procedure.

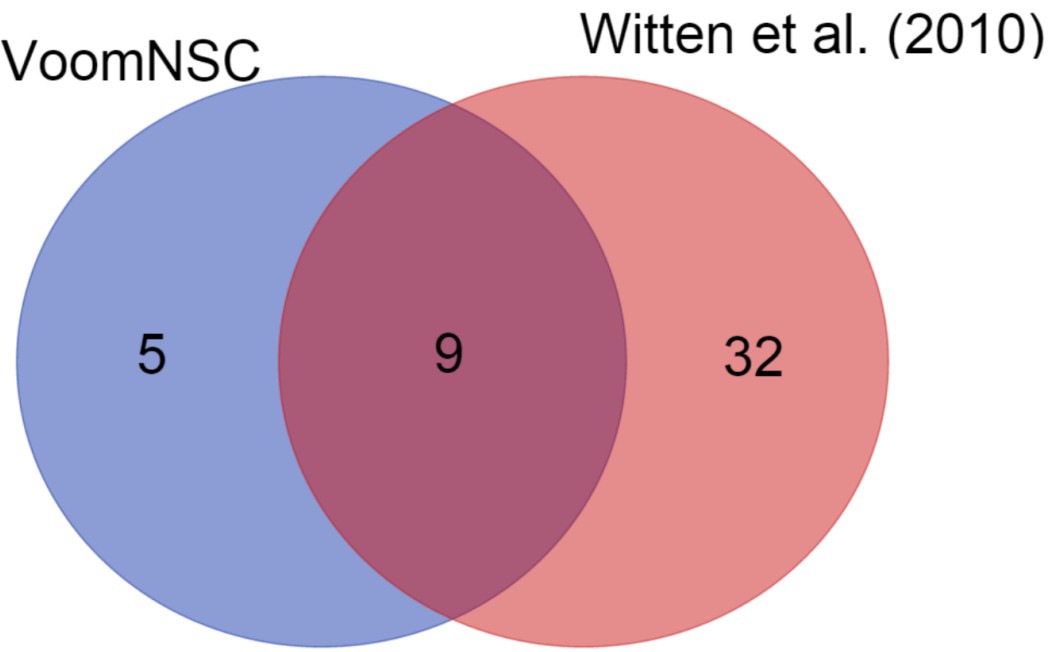

**Figure 6  A Venn-diagram displaying the number of selected miRNAs.**

## CONCLUSION

We conclude that the PLDA algorithm with power transformation and voomNSC classifiers may be the sparse methods of choice, if one aims to obtain accurate models for RNA-Seq classification. SVM and RF algorithms are the overall winners in nonsparse classifiers. When sparsity is the measure of interest, voomNSC classifiers should be the preferred methods. Along with its accurate and sparse performance, the voomNSC method is fast and applicable to even very large RNA-Seq datasets. Besides the prediction purpose, the voomNSC classifier can be used to identify the potential diagnostic biomarkers for a condition of interest. In this way, a small subset of genes, which is relevant to distinguishing the different classes, can be detected. These genes can then be investigated for further, such as discovering additional genes which have interactions with these genes. We leave extending this model with considering the known biomarkers as a follow-up research study.

We believe that this study may contribute to other studies in proposing the voom extensions of powerful statistical learning classifiers including SVM, RF, etc. We also recommend extending this approach for other types of statistical analysis methods such as clustering analysis. These generalizations may allow users to analyze both microarray and RNA-Seq data with similar workflows and provide comparable results. For the applicability of the proposed approaches, we developed a user-friendly and easy-to-use web-based tool. This tool can be accessed through http://www.biosoft.hacettepe.edu.tr/voomDDA/.

**List of abbreviations**

| | |
|---|---|
| **voom** | Variance modeling at observational level |
| **NSC** | Nearest shrunken centroids |
| **RNA** | Ribonuclic acid |
| **NGS** | Next-generation sequencing |
| **PLDA** | Poisson linear discriminant analysis |
| **NBLDA** | Negative binomial linear discriminant analysis |
| **DE** | Differential expression |
| **limma** | models for microarray and RNA-Seq data |
| **DDA** | Diagonal discriminant analysis |
| **DLDA** | Diagonal linear discriminant analysis |
| **DQDA** | Diagonal quadratic discriminant analysis |
| **log-cpm** | log counts per million |
| **SAM** | significance analysis of microarrays |
| **vst** | Variance stabilizing transformation |
| **rlog** | Regularized logarithmic transformation |
| **SVM** | Support vector machines |
| **RF** | Random forests |
| **miRNA** | micro-RNA |
| **TMM** | Trimmed mean of M values |
| **NB** | Negative binomial |
| **ADAS-Cog** | Alzheimer Disease Assessment Scale-cognitive subscale |
| **WMS** | Wechsler Memory Scale |
| **MMSE** | Mini-Mental State Exam |
| **CDR** | Clinical Dementia Rating |
| **RCC** | Renal cell cancer |
| **TCGA** | The Cancer Genome Atlas |
| **KIRP** | Kidney renal papillary cell |
| **KIRC** | Kidney renal clear cell |
| **KICH** | Kidney chromophobe carcinomas |
| **LUAD** | lung adenocarcinoma |
| **LUSC** | lung squamous cell with carcinoma |

## ACKNOWLEDGEMENTS

We would like to thank A. Keller for sharing the Alzheimer data, also thank W. Huber, S. Anders and M.I. Love for insightful discussions on the concept of the algorithm and simulation settings of this paper.

### Funding

This work was supported by the Research Fund of Erciyes University [TDK-2015-5468]. The funders had no role in study design, data collection and analysis, decision to publish, or preparation of the manuscript.

## Grant Disclosures

The following grant information was disclosed by the authors:
Research Fund of Erciyes University: TDK-2015-5468.

## Competing Interests

The authors declare there are no competing interests.

## Author Contributions

- Gokmen Zararsiz conceived and designed the experiments, performed the experiments, analyzed the data, contributed reagents/materials/analysis tools, wrote the paper, prepared figures and/or tables.
- Dincer Goksuluk conceived and designed the experiments, performed the experiments, analyzed the data, contributed reagents/materials/analysis tools, prepared figures and/or tables.
- Bernd Klaus conceived and designed the experiments, contributed reagents/materials/-analysis tools.
- Selcuk Korkmaz conceived and designed the experiments, performed the experiments, analyzed the data, contributed reagents/materials/analysis tools.
- Vahap Eldem performed the experiments, analyzed the data, wrote the paper.
- Erdem Karabulut and Ahmet Ozturk wrote the paper, reviewed drafts of the paper.

## Data Availability

The raw data is available in the Supplemental Files, and the code repository can be found at https://github.com/gokmenzararsiz/voomDDA.

## Supplemental Information

Supplemental information for this article can be found online at http://dx.doi.org/10.7717/peerj.3890#supplemental-information.

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
