# Peer review of "voomDDA: discovery of diagnostic biomarkers and classification of RNA-seq data"

_PeerJ, doi:10.7717/peerj.3890_

## Round 0.1 · original submission · Major Revisions

Dear Gokmen,

We now have three reports from the reviewers who have made a number of comments as well as suggests. In particular, two reviewers raised the issue of "validity of the findings" and I would suggest that you explain that how you split the datasets and the results.

Finally I should mention that PeerJ is an Open Access journal for all the readers and any software packages developed with our published paper should be open access in the public domain.

I look forward to receiving your revised version in the near future.

Kind regards,

Zemin Ning

Reviewer 1 ·

Basic reporting

In summary, this study was well planned, efficiently organized, and competently reported. The manuscipt presentation and literacy standard are generally high.

Experimental design

Some of the real RNA-sequencing datasets were download from public database. They may produce from different laboratories. Measurements are affected by laboratory conditions, reagent lots and personnel differences. Did the author takes the batch effect into account? Will this affect your predicted results?

Validity of the findings

In evaluation process paragraph (line 428-429), the data were randomly split into two parts as training (70%) and test (30%) set. We know that test errors may differ for different train/test splits.(line 436).How dose the user determine the appropriate split ratio?

Reviewer 2 ·

Basic reporting

The manuscript by Zararsiz et al. introduced a method which combined voom and nearest shrunken centroids/diagonal discriminant classifiers together to perform classification of RNA-seq data. The methods are interesting, particularly for its possible application in the discovery of diagnostic biomarkers based on RNA-Seq data. Nevertheless, there are several weaknesses that should be improved. My major concerns and specific comments are listed below.

1) The manuscript was not very well written, containing few typos and formatting/grammar errors. The equations lists in the manuscript seem to be correct but not easy to follow, detailed mathematical annotations are encouraged to be included.

Experimental design

2) The purposed method VoomDLDA/VoomDQDA or VoomNSC didn't show superior performance compared to conventional approaches, in both simulation (Fig3) and real data. It will thus limit the application in practice.

3) There is large paragraph descriping data normalization in the Materials & Methods section. However, the final results showed that expression data from raw read counts, median ratio normalization or TMM normalization are basically identical (Fig3 & Fig4), any explanation for that?

Validity of the findings

4) For the purpose of discovering diagnostic biomarkers, it would be more relevent to train the model to prioritize the known biomarkers. Unfortunately, I'm not ware the author performed analysis of the kind in the manuscript.

5) The author split the data into training and test sets with 70% and 30%, is it part of cross validation, please clarify.

Reviewer 3 ·

Basic reporting

no comment

Experimental design

no comment

Validity of the findings

no comment

Additional comments

The paper introduces a novel data normalization, processing, classification, and feature selection method for RNA-Seq count data, by combining Voom normal linear modeling method with NSC (nearest shrunken centroids) and other classification methods. The presentation and experimentation is comprehensive. The authors utilize both simulated and real datasets and compare their results with some of the commonly used classification methods.

I commend the authors for making their source code available. The method is also made available via a web service.

I do not have any major concerns. I have listed the following points which would improve the publication:

* In the abstract, keep background info to a minimum (RNAseq advantages not necessary to list in abstract) and highligh the contributions more.

* The authors call their method voomNSC in the abstract, but voomDDA in the web link and the title. Use one name for the method, consistently. What does voomDDA stand for? If voomDDA is the umbrella term, the abstract should explain that rather than just voomNSC.

* Expand the Voom acronym in the abstract ("variance modeling at the observational level")

* In intro, authors introduce Voom as a good recently developed method and list its advantages. But a motivation for extending Voom is missing. What does Voom not handle that this paper is addressing?

* SVM seems to perform well, but the main criticism of the authors is its inability to produce sparse models. SVM is frequently applied in combination with a feature selection strategy (e.g., forward selection or backward elimination). Performing SVM without feature selection is an unfair comparison.

* In Equation 1: what is X.i ?

* In Tables, mark the winners in bold.

* Include other classifiers (e.g., SVM) also in the webtool.

* Network analysis and GO analysis may not be appropriate for the selected classifier genes. Sparse set of genes that are good for classification would mask other coregulated and functionally relevant genes that are necessary for meaningful downstream analysis.

* The web address in pdf is hyperlinked to "http://www.biosoft.hacettepe.edu.tr/voomDDA/.%20". The extra .%20 causes problems.

* Heatmap, Network, and Go features seem to require the user having to run VoomDDA analysis first. Disable Heatmap/Network/Go until VoomDDA is executed, or automatically execute it.

* Proofread the paper for syntax and writing errors. Some errors are:
* counts which [are] obtained from
less advancements: awkward phrasing
we benefit from the delta rule: replace "benefit"
lead to obtain[ing]
thenumber, usingdeseq, parameterthat, thenormal, miRNAswith, andlung, isapplied

---

## Round 0.2 · accepted · Accept

I take the pleasure to accept your paper and many congratulations!

Reviewer 1 ·

Basic reporting

no comment

Experimental design

no comment

Validity of the findings

no comment

Additional comments

All my concerns were well answered.

Reviewer 2 ·

Basic reporting

The authors have largely addressed my comments. The revised manuscript is better and I am happy with the content of this manuscript.

Experimental design

As above.

Validity of the findings

As above.

Additional comments

As above.